# Phenanthrene Amplifies Microcystin-Induced Toxicity in the Submerged Macrophyte *Vallisneria natans*

**DOI:** 10.3390/toxins17090472

**Published:** 2025-09-21

**Authors:** Xiang Wan, Yi Zhang, Yucong Li, Fei Yang, Liqiang Xie

**Affiliations:** 1College of Environment and Ecology, Jiangsu Open University, Nanjing 210017, China; 2Jiangsu Engineering and Technology Centre for Ecological and Environmental Protection in Urban and Rural Water Environment Management and Low Carbon Development, Nanjing 210017, China; 3State Key Laboratory of Lake Science and Environment, Nanjing Institute of Geography and Limnology, Chinese Academy of Sciences, 73 East Beijing Road, Nanjing 210008, China; 4Nanjing Institute of Environmental Sciences, Ministry of Ecology and Environment of the People’s Republic of China, Nanjing 210042, China

**Keywords:** cyanotoxins, polycyclic aromatic hydrocarbons, synergetic effect, photosynthetic fluorescence

## Abstract

Microcystin–LR (MC-LR) and phenanthrene (Phen), which commonly co-occur in eutrophic waters, have been extensively studied as individual contaminants, but their combined ecotoxicological effects on submerged macrophytes remain unclear. In this study, we examined the individual and combined toxicity of MC-LR (2, 10, 50, 250, and 1000 μg/L) and Phen (0.2, 1, 5, 25, and 100 μg/L) on the submerged macrophyte *Vallisneria natans* over a 7-day exposure. Key toxicity biomarkers, including growth, photosynthetic efficiency, and antioxidant responses (catalase, superoxide dismutase, glutathione S-transferase, and malondialdehyde), were evaluated. The results showed that high concentrations of each contaminant alone (MC-LR ≥ 1000 μg/L; Phen ≥ 100 μg/L) significantly inhibited growth and reduced photosynthetic efficiency. In contrast, synergistic toxicity was observed at much lower combined concentrations (≥50 + 5 μg/L), with effects substantially exceeding those of individual exposures. Co-exposure intensified antioxidant activity, but it was insufficient to mitigate oxidative damage. Notably, Phen at concentrations above 25 μg/L significantly enhanced the bioaccumulation of MC-LR in *V. natans*. These findings demonstrate that environmentally relevant mixtures of MC-LR and Phen induce remarkable toxicity even at concentrations where individual compounds show negligible effects. The results highlight that co-existing cyanotoxins and polycyclic aromatic hydrocarbons may present greater ecological risks than predicted from single-contaminant assessments, underscoring the need to update current ecological risk frameworks for the accurate evaluation of complex pollution scenarios in freshwater systems.

## 1. Introduction

Increases in surface water temperatures and frequent heatwaves under climate change have accelerated the global spread of harmful cyanobacterial blooms in freshwater ecosystems [1,2]. These toxic algal proliferations and their associated toxins pose significant threats to both aquatic ecosystem stability and human health [3]. Among cyanotoxins, microcystins (MCs) represent the most prevalent class and are produced by various species of freshwater planktonic cyanobacteria, such as *Microcystis*, *Dolichospermum*, *Aphanizomenon*, *Planktothrix*, and *Nostoc* [4]. As potent hepatotoxins, MCs exert deleterious effects on aquatic biota and human health through acute and chronic exposure pathways [4,5]. To date, more than two hundred structural MC variants have been identified [4]. A provisional guideline limit of 1 μg/L for MC-LR in drinking water has been established by the WHO, targeting this most toxic and studied congener [6]. However, environmental MC concentrations frequently exceed this safety threshold, occasionally resulting in drinking water crises [3,7]. A notable example occurred in 2014 when approximately 500,000 residents in the Lake Erie watershed in the USA were advised against drinking tap water due to MC levels in treated drinking water surpassing the safe limits [7].

Although eutrophication and elevated temperatures are well-characterized drivers of cyanobacterial blooms [1,3,8,9], recent studies indicate that organic pollutants, including polycyclic aromatic hydrocarbons (PAHs), antibiotics, and other emerging contaminants at environmentally relevant concentrations, can exacerbate bloom intensity by stimulating cyanobacterial growth and enhancing MC production in eutrophic waters [10]. The frequent co-occurrence of MCs and organic contaminants in freshwater systems highlights a critical need to evaluate their combined toxicity, especially considering their potential synergistic effects on aquatic ecosystem integrity [11].

PAHs, a concerning class of persistent organic pollutants originating from incomplete combustion processes, are ubiquitously distributed in aquatic environments [12]. These compounds are consistently detected at elevated concentrations in aquatic ecosystems and demonstrate significant bioaccumulation potential in aquatic organisms [13]. On a global scale, the concentration of PAHs in lakes across China is moderately high, with ∑PAHs levels in water and sediment ranging from 4.0 to 12,970.8 ng/L and 6.52 to 7935.21 ng/g, respectively [13]. Their environmental persistence, long-range transport capacity, and carcinogenicity have raised significant global concern [12,13]. Among PAHs, phenanthrene (Phen) has emerged as a priority research target owing to its high detection frequency in freshwater systems, with reported porewater concentrations exceeding 20 μg/L in contaminated environments [14]. These characteristics establish Phen as an ideal model compound for investigating organic pollutant dynamics and ecological impacts [5,15].

Freshwater ecosystems face growing threats from the synergistic effects of MC and PAH co-contamination, particularly in eutrophic systems [5,15]. Although ecological risks of such combined pollution are gaining recognition, the combined toxic toxicity of MCs and PAHs, especially their impacts on submerged macrophytes, remains poorly understood [15,16]. These foundation species play vital roles in freshwater ecosystems, yet their global decline has been associated with the dual stressors of harmful algal blooms and complex pollution in eutrophic waters [17]. To address these knowledge gaps. *Vallisneria natans* (Lour.), a cosmopolitan submerged macrophyte [18], was used as a test organism to evaluate the individual and combined toxic effects of MC-LR and Phen. The study objectives were as follows: (1) examine the growth responses and chlorophyll photosynthetic performance of *V. natans* under single or combined exposures; (2) characterize oxidative stress responses through key biomarkers, including superoxide dismutase (SOD), catalase (CAT), glutathione S-transferase (GST), and malondialdehyde (MDA) of *V. natans*; and (3) assess MC-LR accumulation in *V. natans* during pollutant interactions. These findings advance ecological risk assessment for eutrophic waters experiencing concurrent harmful algal blooms and organic contamination.

## 2. Results

### 2.1. Growth and Photosynthetic Responses

The effects of varying concentrations of MC-LR and Phen on *V. natans* growth are illustrated in Figure 1A–C. After seven days of exposure, MC-LR alone had no significant influences on root length or total fresh weight, compared with the control (without any toxicants). Significant inhibition (*p* < 0.05) in shoot length was only observed at MC-LR concentrations ≥ 1000 μg/L compared to controls (Figure 1A). Low concentrations of Phen (≤5 μg/L) slightly stimulated growth in *V. natans*, but all growth parameters declined significantly (*p* < 0.05) at 100 μg/L Phen. Root length proved the most sensitive indicator, showing significant inhibition (*p* < 0.05) at a lower concentration of Phen (25 μg/L) (Figure 1B). Under combined exposure to MC-LR and Phen, *V. natans* growth suppression intensified with increasing mixture concentrations. Both shoot length and total fresh weight were significantly (*p* < 0.05) lower than that in controls and their individual exposure treatments at mixture concentrations ≥ 250 μg/L MC-LR + 25 μg/L Phen (Figure 1A,C). Root length responded more rapidly, declining significantly (*p* < 0.05) at 50 μg/L MC-LR + 5 μg/L Phen (Figure 1B).

As shown in Figure 1D, low concentrations of MC-LR (≤10 μg/L) slightly enhanced the photosynthetic efficiency (*Fv/Fm*) of *V. natans* relative to controls. In contrast, almost no stimulatory effect was observed when Phen applied alone. Although the highest individual concentrations of either contaminant induced visible but non-significant (*p* > 0.05) decreases in *Fv/Fm*, the MC-LR + Phen mixture (≥ 250 + 25 μg/L) caused significant reductions compared to controls (*p* < 0.05). Notably, at the highest concentration, the mixture also induced a significant decrease in *Fv/Fm* compared to exposure to either contaminant alone.

### 2.2. Oxidative Response

Figure 2 demonstrates that 7-day exposure to high concentrations of MC-LR (≥250 μg/L), co-exposure to MC-LR + Phen (≥50 + 5 μg/L), and 100 μg/L Phen significantly induced SOD activity in *V. natans* (*p* < 0.05 vs. controls). Furthermore, the combined MC-LR + Phen exposure significantly (*p* < 0.05) elevated SOD activity relative to exposure to Phen alone (≥5 μg/L). However, only the MC-LR + Phen mixture at 250 + 25 μg/L significantly (*p* < 0.05) increased SOD activity relative to exposure to 250 μg/L MC-LR alone (Figure 2A). CAT activity generally exhibited a response trend similar to SOD across the different toxicant treatments. The key difference was that significant (*p* < 0.05) enhancement of CAT activity by the mixtures occurred only at higher Phen concentrations (≥25 μg/L) within the mixtures (Figure 2B). GST activity showed no significant response to Phen (0.1–100 μg/L) alone, but increased significantly at high concentrations of MC-LR (≥250 μg/L) and MC-LR + Phen (≥250 + 25 μg/L). At the highest exposure level, the mixture induced a significantly (*p* < 0.05) greater increase in GST activity than either contaminant alone (Figure 2C). MDA content gradually increased with rising concentrations of both single and combined exposures. Significant increases (*p* < 0.05) in MDA occurred only at the highest concentrations of MC-LR or Phen alone. In contrast, the mixture caused a significant (*p* < 0.05) increase in MDA already at the second-highest concentration tested (Figure 2D).

### 2.3. Accumulation

The accumulation of MC-LR in *V. natans* seedlings exposed to a single MC-LR or MC-LR + Phen mixture is shown in Figure 3. MC-LR content increased with exposure concentration for both treatments after seven days. At higher mixture concentrations (≥250 μg/L MC-LR + 25 μg/L Phen), MC-LR accumulation was significantly (*p* < 0.05) greater than that observed with exposure to the corresponding concentration of MC-LR alone. However, at lower concentrations (≤50 μg/L MC-LR and ≤50 + 5 μg/L MC-LR + Phen), MC-LR content showed no significant difference between plants exposed to MC-LR alone or in combination with Phen.

### 2.4. Assessment

Root length, total fresh weight, and *Fv/Fm* were employed as toxicity indicators to assess the combined toxicity of MC-LR and Phen on *V. natans* seedlings (Table 1). The calculated inhibition ratios (*RI*) for all three indicators revealed a consistent pattern. At lower MC-LR + Phen mixture concentrations (≤10 + 1 μg/L), *RI* values were significantly less than 1 (*p* < 0.05), indicating an antagonistic interaction between the two toxicants. Conversely, at higher mixture concentrations (≥50 + 5 μg/L), *RI* values were significantly greater than 1 (*p* < 0.05), demonstrating a synergistic interaction. Notably, the maximum *RI* values occurred at an intermediate mixture concentration (250 + 25 μg/L), followed by a gradual decline at higher concentrations, suggesting a dose-dependent attenuation of the synergistic effect.

## 3. Discussion

*V. natans* is a perennial, submerged clonal macrophyte widely found in aquatic ecosystems across the globe. Due to its strong adaptability and the ease of obtaining seedlings from seeds, *V. natans* is frequently employed as a model species for studying the toxicological effects of pollutants in aquatic environments [18]. Submerged macrophytes are vital components of freshwater ecosystems, with *V. natans* serving as a key pioneer species [19]. However, increasing evidence suggests that the decline in submerged macrophytes may be associated with toxic algal blooms and complex pollution in eutrophic waters [10,17]. While previous studies have predominantly focused on the effects of individual biotic and abiotic stressors during harmful algal blooms on submerged plants, investigations into combined stressor interactions remain limited [11,15,20]. Given that multiple pollutants often coexist in natural aquatic environments, this study investigates the combined effects of MC-LR and Phen on *V. natans* to better understand their ecological interactions.

Previous studies have demonstrated that MCs can accumulate in plant tissues, causing adverse effects such as growth inhibition, ultrastructure damage, and reduced photosynthetic efficiency [20,21]. However, our results revealed that *V. natans* exhibited remarkable tolerance to MC-LR at environmentally relevant concentrations, with no significant growth impairment observed. These findings align with prior research by Wang et al. [21], which reported that *V. natans* seedlings remain unaffected at low MC-LR concentrations (0–1 mg/L), while higher doses (≥4 mg/L) significantly suppress growth after prolonged exposure (4–8 days). Research on the toxicological effects of PAHs on submerged macrophytes remains limited, as existing studies on aquatic plants have predominantly employed duckweed and algae as model organisms [22,23]. In our previous study, Phen below 10 μg/L had no significant adverse effect on *Lemna gibba* growth [15]. Furthermore, Ren et al. [22] reported fluoranthene and pyrene toxicity thresholds of 0.5 mg/L and 50% growth inhibition at 16 mg/L for this species. Our findings further reveal a high tolerance in *V. natans* to Phen alone at environmentally relevant concentrations (≤5 μg/L), which may explain its suitability for PAHs bioremediation [24]. Notably, the roots of *V. natans* exhibited greater sensitivity than shoot length metrics, developing necrosis at elevated Phen concentrations (≥100 μg/L). This aligns with established patterns of PAH accumulation primarily in root tissues [25]. Root systems serve as the primary uptake site for nutrients and minerals in submerged macrophytes [26]. However, high-concentration PAHs form hydrophobic root-surface films that significantly impede water and mineral nutrient uptake, ultimately suppressing growth and biomass accumulation [27].

The chlorophyll fluorescence parameter *Fv/Fm*, representing the maximum quantum yield of photosystem II (PSII) after dark-adaptation, serves as a sensitive indicator of photosynthetic performance [28]. Interestingly, a hormetic response in photosynthetic efficiency was observed at low MC-LR concentrations (≤10 μg/L), consistent with findings by Cheng et al. [18], which stated that *Fv/Fm* values in *V. natans*, exposed to MC-LR at similar concentrations, fell within the normal range of plants. In contrast, inhibitory effects were observed at higher concentrations. This biphasic dose-response pattern aligns with documented effects in other aquatic macrophytes (e.g., duckweeds) under MC-LR exposure [21]. Furthermore, Phen elicited concentration-dependent reductions in *Fv/Fm*, with maximal suppression observed at the highest concentration tested. This finding corresponds with previous reports of chlorophyll decreases in terrestrial and aquatic plants exposed to elevated PAHs levels [16,29,30]. For instance, Yang et al. [16] observed a marked decrease in chlorophyll-a content in a duckweed exposed to naphthalene concentrations exceeding 10 μg/L. Ahammed et al. [30] reported reduced *Fv/Fm* values in five vegetable crops (pakchoi, cucumber, cabbage, tomato and lettuce) exposed to Phen at 30–300 μM. Collectively, these findings suggested that elevated PAH levels can negatively affected photosynthesis in aquatic plants.

Pollutant exposure and other stressors can induce molecular damage directly or through reactive oxygen species (ROS) formation [31]. Organisms mitigate ROS via evolved antioxidant systems, including enzymatic (e.g., SOD, CAT, GST) and non-enzymatic components like glutathione [32]. SOD primarily catalyzes the dismutation of •O_2_^−^ to O_2_ and H_2_O_2_, which CAT subsequently detoxifies [33]. When ROS production overwhelms antioxidant capacity, lipid peroxidation (measured as MDA) increases significantly [21]. Multiple studies confirm that PAHs activate antioxidant responses in both terrestrial and aquatic plants [30,34]. For examples, Yin et al. [34] observed elevated •O_2_^−^ production and increased activities of SOD, CAT and GST in the submerged macrophyte *Ceratophyllum demersum* L. exposed to pyrene (10–100 μg/L). Similarly, Ahammed et al. [30] reported induced SOD and CAT activities alongside elevated MDA content in five vegetable crops under Phen exposure (30–300 μM). Studies also demonstrate antioxidant enzyme induction in submerged macrophytes exposed to MCs [21,35]. Notably, GST may mediate the initial detoxification step in aquatic organisms by conjugating MCs with glutathione, resulting in a less toxic conjugate than MCs alone [36]. Consistent with this detoxification role, our results revealed a more pronounced GST response in seedlings exposed to MC-LR compared to those exposed to Phen. Furthermore, exposure to MC-LR, Phen, and their mixture generally induced significant elevations in antioxidant enzyme activities and MDA levels. This effect was particularly evident at higher individual concentrations and during combined exposures (Figure 2). Collectively, these findings indicate that toxicants exposure triggered oxidative stress, and that the antioxidant defense systems were activated to mitigate the damage caused by these toxicants.

Few investigations have systematically assessed the toxicological effects of simultaneous MCs and co-contaminant exposure in aquatic plant systems [11]. Our results demonstrate that MC-LR and Phen exert combined toxicity on *V. natans*, exhibiting antagonistic effects at low concentrations but synergistic effects at high concentrations (based on *RI* values). The synergistic interactions were characterized by enhanced bioaccumulation of MC-LR (Figure 3), severe growth retardation and photosynthetic inhibition, and exaggerated antioxidant responses (Figure 1 and Figure 2), compared to individual exposures. Based on current literature [16,21], the following mechanisms for the MC-LR/Phen synergism were proposed (Figure 4). First, interactions occur extracellularly prior to MC-LR cellular uptake. Co-exposure likely increases membrane permeability, facilitating greater MC-LR accumulation in plant tissues. Similar phenomenon was observed in aquatic plants exposed to MC-LR with PAHs (e.g., naphthalene and Phen) [15,16], surfactants [37], and other emerging pollutants [11]. Second, elevated intracellular concentrations of toxicants in chloroplasts synergistically disrupt photosynthetic machinery, leading to massive superoxide (O_2_•^−^) generation [21]. Third, the combined toxicant stress surpasses antioxidant defenses, leading to overwhelmed ROS scavenging capacity and resulting in complete growth cessation followed by cellular necrosis [16].

Our findings demonstrate that MC-LR and Phen induce synergistic toxicity at environmentally relevant concentrations (≥50 μg/L MC-LR + ≥5 μg/L Phen), consistent with previous observations in duckweed (*Lemna gibba*) systems [15]. Notably, these concentrations correspond to documented pollution levels in real aquatic environments (e.g., eutrophic lakes receiving agricultural and industrial runoff) [14,38]. For instance, dissolved MC concentrations can exceed 2 mg/L during cyanobacterial bloom collapse events [38]. Furthermore, Phen concentrations of 2.44–26.1 μg/L have been measured in river estuary porewater [14]. Such evidence confirms that these synergistic interactions are likely to occur in natural environments. Consequently, co-occurrence of these two pollutants may trigger abrupt vegetation loss in contaminated habitats, particularly for foundation species like *V. natans*. Moreover, the limited adaptive capacity to combined stressors implies severely impaired bioremediation potential in multi-pollutant scenarios. These collective insights highlight an urgent need to revise ecological risk assessment frameworks by incorporating contaminant interactions, thereby protecting keystone species in eutrophic freshwaters experiencing concurrent cyanobacterial blooms and complex organic pollution.

## 4. Conclusions

This study systematically evaluated the individual and joint effects of MC-LR and Phen on *V. natans*. Although neither contaminant alone induced significant phytotoxicity except at the highest concentrations, their combined exposure at environmentally relevant concentrations (MC-LR + Phen ≥ 50 + 5 μg/L) resulted in synergistic interactions. Combined exposure markedly enhanced MC-LR bioaccumulation, severely inhibited growth, impaired photosynthetic efficiency, and provoked exaggerated antioxidant responses. Mechanistically, we propose that combined pollutant exposure overwhelms physiological defense systems through amplified toxicant accumulation, ultimately disrupting photosynthetic function and antioxidant capacity. These findings suggest that aquatic ecosystems with co-occurring MC and PAH contamination are particularly vulnerable to synergistic effects. Consequently, environmental management strategies must consider the combined impacts of such multiple stressors. This study highlights the necessity for comprehensive ecological risk assessments of co-occurring cyanotoxins and PAHs.

## 5. Materials and Methods

### 5.1. Experimental Materials

In this study, *V. natans* seeds were sourced commercially. Phen (CAS No.: 85-01-8, purity ≥ 99%) was purchased from Sigma-Aldrich (St. Louis, MO, USA), while MC-LR (CAS No.: 101043-37-2, purity ≥ 95%) was supplied by Taiwan Algal Science Inc. All chemicals were stored at −20 °C until use. For MC-LR extraction and analysis, HPLC-grade methanol (Tedia Company, Fairfield, OH, USA) was employed. All other reagents used were of analytical grade purity.

### 5.2. Experimental Design

Healthy mature *V. natans* seeds were selected for seedling cultivation. Prior to germination, seeds underwent sterilization in 1.0% (*w*/*v*) sodium hypochlorite solution for 10 min, followed by three rinses with sterile distilled water. Sterilized seeds were then cultivated in 1/10 Hoagland nutrient solution [39], under a light intensity of 45 μmol m^−2^ s^−1^, with a 14-h light/ 10-h dark cycles at 25 °C. After 14 days of growth, healthy and uniform seedlings were chosen and cleaned in preparation for the subsequent experiments.

For the experiment, forty seedlings were placed in each 100 mL conical flask containing 60 mL of sterilized 1/10 Hoagland solution. The flasks were set up with varying levels of toxicants (MC-LR, Phen, or their combination) or without toxicants (as a control). The concentration ranges for MC-LR concentrations were 0, 2, 10, 50, 250 and 1000 μg/L, and for Phen, they were 0, 0.2, 1, 5, 25, 100 μg/L. It has been reported that dissolved MC concentrations can exceed 2 mg/L during cyanobacterial bloom collapse events [38]. Meanwhile, Phen concentrations measured in river estuary porewater have been found to range from 2.44 to 26.1 μg/L [14]. To reflect these environmentally relevant concentrations, the chosen ranges in this study were selected to encompass levels commonly encountered in contaminated aquatic systems. The experiment was maintained for 7 days in a growth chamber under identical environmental conditions to seedling cultivation, with four biological replicates per treatment. During the experiment, the medium was renewed every two days to maintain relatively consistent chemical and nutrient concentrations. After 7 days of exposure, growth parameters (root/shoot length, fresh weight), chlorophyll fluorescence (*Fv/Fm*) and biochemical parameters (SOD, CAT, GST, MDA) of seedings from different treatments were measured. Additionally, the accumulation of MC-LR in the seedlings was determined.

### 5.3. Analytical Methods

#### 5.3.1. Determination of Growth Parameters and Photosynthetic Efficiency (*Fv/Fm*)

Growth parameters including shoot/root length and total weight were measured by randomly selecting ten seedlings from each replicate flake. Shoot length was determined by measuring the length of the longest leaf, while root length was determined by measuring the primary root, using a ruler with an accuracy of 1 mm. Total fresh weight was recorded using an electronic analytical balance with an accuracy of 0.1 mg. Photosynthetic efficiency was assessed by measuring the dark-adapted fluorescence ratio *Fv/Fm*, a well-established indicator of PSII photochemical efficiency [28]. *Fv/Fm* measurements of the leaves were determined using a portable plant efficiency analyzer (Handy PEA, Hansatech, Norfolk, UK). Prior to measurement, leaves were dark-adapted for 20 min, followed by a 1-s exposure to saturating light (1500 μmol m^−2^ s^−1^ at 650 nm peak wavelength) [18].

#### 5.3.2. Determination of Biochemical Indicators

*V. natans* seedling samples (0.1 g fresh weight) after experimental period were homogenized in 2.0 mL ice-cold potassium phosphate buffer (100 mM, pH 7.0) using a pre-chilled mortar and pestle. The homogenates were then centrifuged at 10,000× *g* for 10 min at 4 °C. The resulting supernatant was collected and stored at −80 °C pending biochemical analysis. The activities of SOD, CAT, GST, and MDA content were measured using commercial assay kits (Nanjing Jiancheng Bioengineering Institute, Nanjing, China) following manufacturer protocols. These included the SOD assay kit (Hydroxylamine method), CAT assay kit (Ammonium molybdate method), GST assay kit (CDNB method), MDA assay kit (TBA method) and total protein quantitative assay kit (BCA method).

#### 5.3.3. MC-LR Extraction and Accumulation Determination

To quantify MC-LR accumulation in *V. natans* seedlings across treatment groups, harvested seedlings were thoroughly (three times) rinsed with sterile distilled water to eliminate any toxicants adhering to the plant surface. MC-LR extraction and analysis followed a previously established protocol [35]. Briefly, fresh plant tissue was homogenized in 2 mL of 70% (*v/v*) methanol, followed by a secondary extraction with 15 mL of 70% methanol. The combined extracts were purified using HLB solid-phase extraction cartridges (Waters, USA). MC-LR quantification was performed on a UHPLC/MS system featuring with an Acquity UPLC BEH C18 column (1.7 mm, 100 mm × 1 mm; Waters, USA). The mobile phase was composed of Solvent A (0.1% formic acid in water) and Solvent B (acetonitrile). The linear gradient was as follows: 0–4.0 min, 75% A; 4.0–4.1 min, 55% A; 4.1–7.0 min, 75% A. Flow rate was 0.5 mL/min with an injected volume of 10 μL. The limit of detection was set at 1 ng/mL (signal to noise S/N 43). Blanks and standards were analyzed after every tenth sample to ensure data quality.

### 5.4. Statistical Analysis

The combined toxicity of MC-LR and Phen were estimated using Abott’s formula, which is a widely used model for evaluating binary contaminant interactions [40]. This model calculates the expected inhibition (*EI_A+B_*) of growth and photosynthetic efficiency for mixtures as:(1)*EI_A+B_* = *I_A_* + *I_B_* − *I_A_ I_B_*/100 where *I_A_* and *I_B_* represent the inhibition percentages induced by individual toxicants. The interaction ratio of inhibition (*RI*) for the mixtures was then determined as:(2)*RI* = *OI_A+B_*/*EI_A+B_* with *OI_A+B_* denoting the observed inhibition for the mixture. The interactive effects were classified according to the *RI* value: *RI* < 1 (antagonism), *RI* = 1 (additivity), and *RI* > 1 (synergism) [40].

All data are expressed as mean ± standard deviation (SD). Differences between groups among different toxicants concentration were analyzed by one-way ANOVA using IBM SPSS Statistics 22.0 software (Chicago, IL, USA). MC accumulation difference were assessed with independent samples *t*-test. Statistical significance was set at *p* < 0.05.

## Figures and Tables

**Figure 1 toxins-17-00472-f001:**
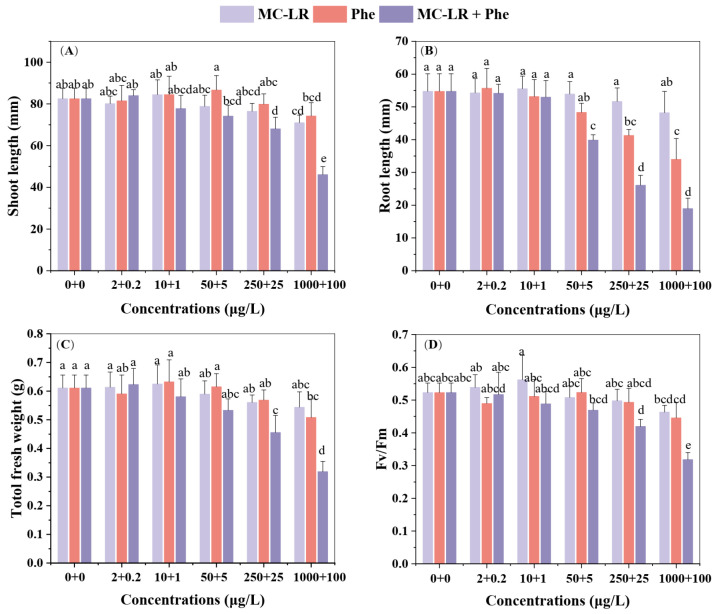
Effects of microcytin–LR (MC-LR) or phenanthrene (Phen) alone and their mixture on growth and photosynthetic efficiency of *V. natans*. (**A**) Shoot length, (**B**) root length, (**C**) total fresh weight, and (**D**) the photosynthetic efficiency parameter (*Fv/Fm*). The data are expressed as mean ± standard deviation (n = 4). Different lowercase letters represent significant differences among the different treatments (*p* < 0.05), as determined by the one-way ANOVA followed by the Duncan test.

**Figure 2 toxins-17-00472-f002:**
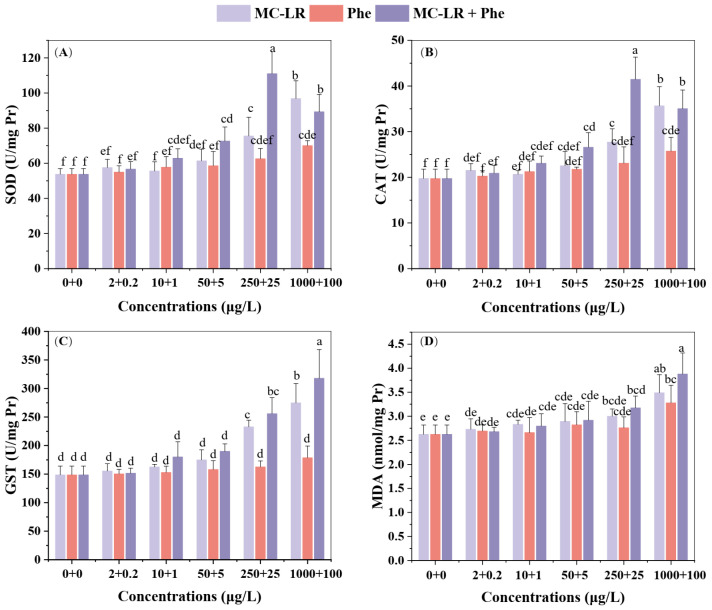
Antioxidant responses of *V. natans* to microcytin–LR (MC-LR) or phenanthrene (Phen) alone and their mixture. (**A**) Superoxide dismutase (SOD) activity, (**B**) catalase (CAT) activity, (**C**) glutathione S transferase (GST) activity, and (**D**) malondialdehyde (MDA) content. The data are expressed as mean ± standard deviation (n = 4). Different lowercase letters represent significant differences among the different treatments (*p* < 0.05), as determined by the one-way ANOVA followed by the Duncan test.

**Figure 3 toxins-17-00472-f003:**
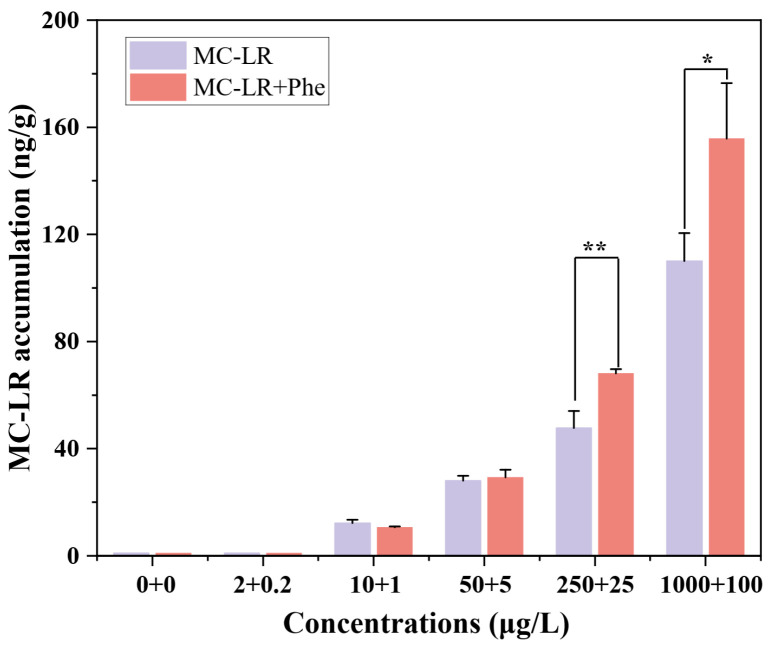
Bioaccumulation of microcystin-LR (MC-LR) in *V. natans* in the presence or absence of phenanthrene (Phen). Values are presented as mean ± standard deviation (n = 4). Asterisks represent statistically significant differences among MC-LR and MC-LR + Phen combined treatments at the same MC-LR concentration, as determined by the Independent-Samples *t*-test. Significant differences are indicated by * (*p* < 0.05) and ** (*p* < 0.01).

**Figure 4 toxins-17-00472-f004:**
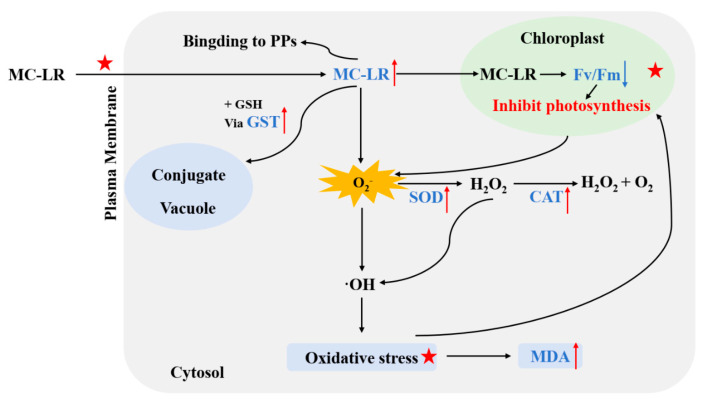
Proposed biochemical response pathway after uptake of MC-LR in *V. natans* cells in the presence of Phen. The processes marked by the red star may involve a potential interaction between MC-LR and Phen. MC-LR: microcystin-LR; Phen: phenanthrene; PPs: protein phosphatases; SOD: superoxide dismutase; CAT: catalase; GST: glutathione S-transferase; MDA: malondialdehyde; H_2_O_2_: hydrogen peroxide.

**Table 1 toxins-17-00472-t001:** The interactive effect of MC-LR and Phen on root length, total fresh weight and *Fv/Fm* of *V. natans*.

Indicators	MC-LR (μg/L)	*I_A_* (%)	Phen (μg/L)	*I_B_* (%)	MC-LR + Phen (μg/L)	*OI_A+B_* (%)	*EI_A+B_* (%)	*RI*	Interactive Effects
Root length	2	0.92 ± 7.93	0.2	−1.65 ± 11.09	2 + 0.2	1.16 + 4.99	−0.73 + 12.06	−1.65	Antagonism
	10	−1.37 ± 6.99	1	2.96 ± 9.44	10 + 1	3.32 + 9.24	1.59 + 7.50	−2.07	Antagonism
	50	1.59 ± 6.86	5	11.79 ± 5.04	50 + 5	27.26 + 2.97	13.38 + 6.12	2.03 *	Synergism
	250	5.73 ± 7.53	25	24.65 ± 3.23	250 + 25	52.36 + 5.45	30.37 + 9.46	1.72 *	Synergism
	1000	12.03 ± 11.86	100	37.96 ± 11.56	1000 + 100	65.36 + 5.77	49.95 + 6.12	1.31 *	Synergism
Total fresh weight	2	−0.59 ± 8.66	0.2	3.22 ± 10.79	2 + 0.2	−2.13 + 9.22	2.63 + 17.69	−0.82	Antagonism
	10	−2.46 ± 10.91	1	−3.70 ± 12.55	10 + 1	4.84 + 10.15	−6.16 + 22.17	−0.78	Antagonism
	50	3.36 ± 7.59	5	−0.89 ± 7.44	50 + 5	12.60 + 6.49	2.46 + 13.03	5.04 *	Synergism
	250	8.13 ± 4.25	25	6.72 ± 5.75	250 + 25	25.33 + 9.74	14.84 + 6.49	1.71 *	Synergism
	1000	10.87 ± 8.79	100	16.71 ± 10.51	1000 + 100	47.69 + 5.82	27.56 + 14.66	1.73 *	Synergism
*F_v_/F_m_*	2	−3.55 ± 7.42	0.2	5.81 ± 3.45	2 + 0.2	0.58 + 13.07	2.26 + 10.71	0.25	Antagonism
	10	−8.07 ± 14.34	1	1.73 ± 10.00	10 + 1	6.02 + 7.25	−6.34 + 17.06	−0.96	Antagonism
	50	2.31 ± 6.74	5	−0.68 ± 8.08	50 + 5	9.76 + 4.21	1.63 + 12.98	6.10 *	Synergism
	250	4.23 ± 6.74	25	5.09 ± 8.08	250 + 25	19.37 + 4.21	9.32 + 12.98	2.08 *	Synergism
	1000	10.93 ± 3.99	100	14.23 ± 8.98	1000 + 100	38.80 + 4.12	25.14 + 8.59	1.55 *	Synergism

*I_A_*: observed inhibition (%) caused by MC-LR; *I_B_*: observed inhibition (%) caused by Phen; *OI_A+B_*: observed inhibition (%) of the mixture; *EI_A+B_*: expected inhibition (%) of the mixture; *RI*: the ratio of inhibition. Values are expressed in mean ± standard deviation (n = 4). Significant interactive effects are indicated by * (*p* < 0.05).

## Data Availability

The original contributions presented in this study are included in the article.

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
