# Peer review of "Phenanthrene Amplifies Microcystin-Induced Toxicity in the Submerged Macrophyte Vallisneria natans"

_toxins, 2025, doi:10.3390/toxins17090472_

Round 1
Reviewer 1 Report
Comments and Suggestions for Authors
Phenanthrene Amplifies Microcystin-Induced Toxicity in the Submerged Macrophyte Vallisneria natans
Toxins
The study is an important contribution to the field of ecotoxicology, offering a new findings on the combined toxicity effects of co-occurring aquatic contaminants (Microcystin and Phenanthrene) on the submerged macrophyte, Vallisneria natans.
This is a well-written article with clear hypotheses and objectives, and a well-defined experimental design.
The analysis of various toxicity biomarkers of Vallisneria natans, including growth, photosynthetic efficiency, and antioxidant responses, provides a comprehensive picture of the physiological stress induced by co-exposure of the contaminants studied.
The results are well described in both text and figures.
The discussion is also well-structured, with emphasis on Figure 4, which could serve as the graphic abstract for this article.
By demonstrating that the combined toxicity of these two contaminants is far greater than the sum of their individual effects, the work has provided evidence for the need to update environmental regulations and monitoring strategies.
Some suggestions for minor adjustments follow:
Abstract
Line 6 - There are studies with Daphnia and other macrophytes and aquatic plants describing the ecotoxicological effects of the combination of these compounds, I suggest rewriting this sentence.
Results
Figure 1 Caption – Please, italicize V. natans.
Discussion
Line 183 - I do not consider a 7-8 day period to be a prolonged evaluation of macrophytes, which have slower growth.
Line 216 - Authors should include the crop types here and consider that terrestrial plants are very different systems from aquatic ones.
Materials and Methods
Line 321 – I suggest including after experimental period.
Line 336 - I suggest changing our to previous
Author Response
Comments from reviewer 1#:
The study is an important contribution to the field of ecotoxicology, offering new findings on the combined toxicity effects of co-occurring aquatic contaminants (Microcystin and Phenanthrene) on the submerged macrophyte, Vallisneria natans. This is a well-written article with clear hypotheses and objectives, and a well-defined experimental design. The analysis of various toxicity biomarkers of Vallisneria natans, including growth, photosynthetic efficiency, and antioxidant responses, provides a comprehensive picture of the physiological stress induced by co-exposure of the contaminants studied. The results are well described in both text and figures. The discussion is also well-structured, with emphasis on Figure 4, which could serve as the graphic abstract for this article. By demonstrating that the combined toxicity of these two contaminants is far greater than the sum of their individual effects, the work has provided evidence for the need to update. Some suggestions for minor adjustments follow:
Abstract
- Line 6 - There are studies with Daphnia and other macrophytes and aquatic plants describing the ecotoxicological effects of the combination of these compounds, I suggest rewriting this sentence.
Response: Agree. Therefore, we particularly highlight that research on the combined effects of these two pollutants on submerged macrophytes is still limited (Line 8).
Results
- Figure 1 Caption – Please, italicize V. natans.
Response: Thank you for pointing this out. We have revised it (Line 102) and checked for the same issue in the entire manuscript.
Discussion
- Line 183 - I do not consider a 7-8 day period to be a prolonged evaluation of macrophytes, which have slower growth.
Response: Thanks for your suggestion. We agree that a longer exposure period (e.g., over 14 days) would be beneficial for assessing effects on slower-growing macrophytes. Nevertheless, the 7–8 day exposure period used in this study was selected based on robust responses observed in previous experiments with Vallisneria natans, where significant effects were detected within 6–8 days (Wang et al., 2107; Cao et al., 2019). It is also worth noting that the use of seedlings in this study may allow a shorter-term assessment, as younger plants tend to exhibit faster growth rates than mature individuals.
- Line 216 - Authors should include the crop types here and consider that terrestrial plants are very different systems from aquatic ones.
Response: Thank you for this comment. We have now included the crop types in the revised manuscript (Line 219). We fully acknowledge that terrestrial plants represent very different systems compared to aquatic plants. However, studies specifically examining chlorophyll response in aquatic plants under phenanthrene exposure are still very limited. Therefore, we have opted to retain the discussion regarding chlorophyll changes in terrestrial plants for comparative reference.
Materials and Methods
- Line 321 – I suggest including after experimental period.
Response: Revised (Line 330).
- Line 336 - I suggest changing our to previous
Response: Revised (Line 344).
References:
- Cao, Q.; Wan, X.; Shu, X.B.; Xie, L.Q. Bioaccumulation and detoxication of microcystin-LR in three submerged macrophytes: The important role of glutathione biosynthesis. Chemosphere 2019, 225, 935–942.
- Wang, Z.; Xiao, B.D.; Song, L.R.; Wang, C.B.; Zhang, J.Q. Responses and toxin bioaccumulation in duckweed (Lemna minor) under microcystin-LR, linear alkybenzene sulfonate and their joint stress. J. Hazard Mater. 2012, 229–230, 137–144.

Reviewer 2 Report
Comments and Suggestions for Authors
There are several studies on the individual ecotoxicological effects of cyanotoxins and organic pollutants. However, research on the combined toxicity of MCs and PAHs, particularly their impacts on submerged macrophytes, is still limited and represents an important novel research area. Overall, the manuscript is well organized. From the introduction to the conclusion, the authors clearly present the background, research gap, objectives, methodology (step by step), results, and discussion. I have only a few minor comments: Line 45: Although Lake Erie is well known, please mention the country in which it is located. Line 59: The authors state that PAHs are consistently detected at elevated concentrations in surface waters. Please provide the quantitative ranges and specify the detected locations. Line 193: The authors mention 50% growth inhibition in Microcystis aeruginosa at 4.29 mg/L Phe. It would be better to compare this with aquatic plants rather than cyanobacteria. Line 292: Please provide the composition of the Hoagland solution or include a reference. Figure 2: Why are SOD and CAT activities highest at 250+25 but lower at 1000+100? Please justify these results. Table 1: The minus sign is presented using a hyphen. Please correct this formatting. Concentrations: What is the basis for selecting the concentration combinations of microcystin-LR and phenanthrene (Phe)? Self-citations: Where possible, please remove self-citations as much as possible. There are so many. (e.g., References 5, 6, 12, 16, 19, 25) and try include more recent references. Bioaccumulation mechanism: The study demonstrates that Phe at concentrations above 25 μg/L significantly enhances the bioaccumulation of MC-LR in V. natans. Please discuss the potential mechanisms behind this observation. Additionally, previous studies report that the coexistence of humic acid with heavy metals significantly reduces bioaccumulation and ecotoxicity in aquatic plants. How can these apparently contrasting effects of different compounds on bioaccumulation be explained?
The language use is generally good, but the manuscript should be carefully proofread for typos and formatting errors. I noticed a few instances.
Author Response
Comments from reviewer 2#:
There are several studies on the individual ecotoxicological effects of cyanotoxins and organic pollutants. However, research on the combined toxicity of MCs and PAHs, particularly their impacts on submerged macrophytes, is still limited and represents an important novel research area. Overall, the manuscript is well organized. From the introduction to the conclusion, the authors clearly present the background, research gap, objectives, methodology (step by step), results, and discussion. I have only a few minor comments:
- Line 45: Although Lake Erie is well known, please mention the country in which it is located.
Response: Thanks for pointing this out. We have added the country of Lake Erie (Line 47).
- Line 59: The authors state that PAHs are consistently detected at elevated concentrations in surface waters. Please provide the quantitative ranges and specify the detected locations.
Response: Thanks for your suggestions. We have provided the quantitative ranges and the detected locations in Line 59─62.
- Line 193: The authors mention 50% growth inhibition in Microcystis aeruginosa at 4.29 mg/L Phe. It would be better to compare this with aquatic plants rather than cyanobacteria.
Response: Agree. Research on the toxicological effects of PAHs on submerged macrophytes remains limited, as existing studies on aquatic plants have predominantly employed duckweed and algae as model organisms. In the current version, we have already included comparisons with duckweed. Accordingly, we have removed the comparison with algae to avoid potential confusion and to better align with the focus of this study.
- Line 292: Please provide the composition of the Hoagland solution or include a reference.
Response: The reference for Hoagland solution preparation has been provided (the reference No. 39).
- Figure 2: Why are SOD and CAT activities highest at 250+25 but lower at 1000+100? Please justify these results.
Response: Thanks for this question. The most significant increases in antioxidant enzyme activities (e.g., SOD and CAT) were observed at the intermediate concentration (250+25), rather than at the highest concentration combination (1000+100). This type of response pattern has been reported in several previous studies (Wang et al., 2017; Yang et al., 2023). A possible explanation is that at the highest toxicant levels, the antioxidant defense systems become overwhelmed by the excessive oxidative stress, leading to a decline in enzymatic activities such as those of SOD and CAT (Wang et al., 2017; Yang et al., 2023).
- Table 1: The minus sign is presented using a hyphen. Please correct this formatting.
Response: Thanks for pointing this out. It has been revised (Table 1).
- Concentrations: What is the basis for selecting the concentration combinations of microcystin-LR and phenanthrene (Phe)?
Response: Thanks for your question. It has been reported that dissolved MC concentrations can exceed 2 mg/L during cyanobacterial bloom collapse events (Lahti et al., 1997). Meanwhile, Phe concentrations measured in river estuary porewater have been found to range from 2.44 to 26.1 μg/L (Maskaoui et al., 2002). To reflect these environmentally relevant concentrations, the chosen ranges in this study were selected to encompass levels commonly encountered in contaminated aquatic systems. The basis for selecting the concentration combinations of these two toxicants has been provided in materials and methods (Line 303-308).
- Self-citations: Where possible, please remove self-citations as much as possible. There are so many. (e.g., References 5, 6, 12, 16, 19, 25) and try include more recent references.
Response: Agree. We have removed self-citations (remaining 3) as much as possible.
- Bioaccumulation mechanism: The study demonstrates that Phe at concentrations above 25 μg/L significantly enhances the bioaccumulation of MC-LR in V. natans. Please discuss the potential mechanisms behind this observation.
Response: Thanks for your question. One potential mechanism for the phenomenon is that their co-exposure likely increases membrane permeability, facilitating greater MC-LR accumulation in plant tissues. Furthermore, similar results were also observed in organisms exposed to MCs and other pollutants (e.g., surfactants and other cyanotoxins) (Wang et al., 2012; Contardo-Jara et al., 2015).
- Additionally, previous studies report that the coexistence of humic acid with heavy metals significantly reduces bioaccumulation and ecotoxicity in aquatic plants. How can these apparently contrasting effects of different compounds on bioaccumulation be explained?
Response: Thank you for your question. Humic acid typically contains multiple functional groups (e.g., oxygen- and nitrogen-based groups), which provide metal-binding sites capable of complexing with heavy metals. The resulting high-molecular-weight complexes are difficult for cells to absorb, thereby reducing the bioavailability and ecotoxicity of the heavy metals. Interestingly, Wang et al. (2017) observed the antagonistic effect on the growth of V. natans between MC-LR and copper. Since MC-LR can also act as a type of dissolved organic matter, it may likewise interact with metals and mitigate their toxicity.
- The language use is generally good, but the manuscript should be carefully proofread for typos and formatting errors. I noticed a few instances.
Response: We have checked carefully and revised typos and formatting errors throughout the manuscript.
References:
- Contardo-Jara, V., et al. Single and combined exposure to MC-LR and BMAA confirm suitability of Aegagropila linnaei for use in green liver systems(®)-A case study with cyanobacterial toxins. Toxicol. 2015, 165, 101–108
- Lahti, K., Rapala, J., Färdig, M., Niemelä, M., Sivonen, K. Persistence of cyanobacterial hepatotoxin microcystin-LR in particulate material and dissolved in lake water. Water Res. 1997, 31(5), 1005–1012.
- Maskaoui, K.; Zhou, J.L.; Hong, H.S.; Zhang, Z.L. Contamination by polycyclic aromatic hydrocarbons in the Jiulong River estuary and western Xiamen Sea, China. Pollut. 2002, 118, 109–122.
- Wang, Z.; Zhang, J.Q.; Li, E.H.; Zhang, L.; Wang, X.L.; Song, L.R. Combined toxic effects and mechanisms of microsystin-LR and copper on Vallisneria Natans (Lour.) Hara seedlings. Hazard Mater. 2017, 328, 108–116.
- Wang, Z., et al. Responses and toxin bioaccumulation in duckweed (Lemna minor) under microcystin-LR, linear alkybenzene sulfonate and their joint stress. Hazard Mater. 2012, 229–230, 137–144.
- Yang, Y.X et al. Combined toxic effects of perfluorooctanoic acid and microcystin-LR on submerged macrophytes and biofilms. Hazard Mater. 2023, 459, 132193.

Reviewer 3 Report
Comments and Suggestions for Authors
Dear authors
Happy day
The paper is fine and interesting but still need some improvements.
Kindly find my comments and suggetions.
- You need sharply a clear reference show that the used toxins are from cyanobacteria and not from the industrial and agricultural discharge. Existing of cyanobacteria in water contaminated with industrial and agricultural discharge did not mean that they are responsible for those described toxins. Kindly fix this point.
- The first sentence in the introduction reflects your interest in the climate change, kindly make it appear leader in the text. First focus on the main source of the described toxins then on the factors that increase their existence.
- Again in line 38 you describe the microcystins (MCs), one need to search if those toxins related to this group or not!
- After reading the introduction I could not find a name of a cyanobacteria that can produce one of those toxins or convert a structure to be one of them! Kindly fix.
- You did not make a replica (concerning the conditions) as in line 295 but you made a replica from the used number. If you made a replica kindly show that in the text.
- In the conclusion part Line 372: I found that the existence of the [co-occurring cyanobacterial blooms and] is odd which you conducted in lab toxicity experiment without any existence of a cyanobacterial sp. You might be more specific and say something like: the coexistence of both compounds increases the toxicity and apparently in nature cyanobacterial different species are able to enhance the toxicity process as well.
- Line 373-374 [Consequently, environmental management 373 strategies must address multiple stressors rather than individual contaminants.] I would like to change this sentence because I am sure that they considering the side-effect of multiple stressors.
- Line 07 and 88 [After seven days of exposure, MC-LR alone had no significant in- 86 fluences on root length or total fresh weight.] kindly add ‘comparing with the control (0 µg/ml).
- The discussion and references parts are fine and covered the aim and the finding of this article.
With my pleasure

Author Response
Comments from reviewer 3#:
Dear authors
Happy day
The paper is fine and interesting but still need some improvements.
Kindly find my comments and suggestions.
- You need sharply a clear reference show that the used toxins are from cyanobacteria and not from the industrial and agricultural discharge. Existing of cyanobacteria in water contaminated with industrial and agricultural discharge did not mean that they are responsible for those described toxins. Kindly fix this point.
Response: Thanks for pointing this out. We agree that the origin of the toxins requires clear attribution. The initial sentence in the abstract was potentially misleading and has been removed in the revised manuscript. We would like to clarify that eutrophication and climate change, rather than direct industrial or agricultural discharge, are the primary drivers of harmful cyanobacterial blooms. Microcystins (MCs) including MC-LR, are well-established cyanobacterial metabolites, notably produced by widespread bloom-forming species such as Microcystis aeruginosa (Huang et al., 2019). Several authoritative references (e.g., references 3, 4, and 9) already cited in the manuscript clearly demonstrate that MCs are synthesized by cyanobacteria.
- The first sentence in the introduction reflects your interest in the climate change, kindly make it appear leader in the text. First focus on the main source of the described toxins then on the factors that increase their existence.
Response: Thanks for your suggestions. Growing evidences indicates that climate warming is triggering the occurrence of harmful cyanobacterial blooms in lakes worldwide (Ho et al., 2019), and since MCs are produced by these blooms, climate change has been given prominence in the introduction. However, the specific mechanisms by which climate change influences MC production are not the primary focus of this study. Information regarding the sources of MCs has been added in the revised manuscript (Lines 38–40). Furthermore, factors affecting cyanobacterial blooms and MC production, such as eutrophication, climate warming, and organic pollutant pollution, have been included in the second paragraph (Lines 49–56).
- Again in line 38 you describe the microcystins (MCs), one need to search if those toxins related to this group or not!
Response: Thanks for pointing this out. Information regarding the sources of MCs has been added in the revised manuscript (Lines 38–40). Among cyanotoxins, MCs represent the most prevalent class and are produced by various species of freshwater planktonic cyanobacteria, such as Microcystis, Dolichospermum, Aphanizomenon, Planktothrix, and Nostoc (Huang et al., 2019).
- After reading the introduction I could not find a name of a cyanobacteria that can produce one of those toxins or convert a structure to be one of them! Kindly fix.
Response: Agree. MCs can be produced by various species of freshwater planktonic cyanobacteria, such as Microcystis, Dolichospermum, Aphanizomenon, Planktothrix, and Nostoc (Huang et al., 2019). Information regarding the sources of MCs has been added in the revised manuscript (Lines 38–40).
- You did not make a replica (concerning the conditions) as in line 295 but you made a replica from the used number. If you made a replica kindly show that in the text.
Response: We thank the reviewer for highlighting this lack of clarity. We confirm that the study included four biological replicates (parallel samples) for each treatment, all conducted simultaneously within the same experiment under identical conditions, as described in Lines 308–310. We did not perform an independent repetition of the entire experiment at a different time. The manuscript has been revised to use the precise term "biological replicates" in Line 305 to avoid any potential misunderstanding.
- In the conclusion part Line 372: I found that the existence of the [co-occurring cyanobacterial blooms and] is odd which you conducted in lab toxicity experiment without any existence of a cyanobacterial sp. You might be more specific and say something like: the coexistence of both compounds increases the toxicity and apparently in nature cyanobacterial different species are able to enhance the toxicity process as well.
Response: Agree. We have revised this sentence (Line 380).
- Line 373-374 [Consequently, environmental management 373 strategies must address multiple stressors rather than individual contaminants.] I would like to change this sentence because I am sure that they considering the side-effect of multiple stressors.
Response: Agree. We have revised this sentence (Line 382).
- Line 07 and 88 [After seven days of exposure, MC-LR alone had no significant in- 86 fluences on root length or total fresh weight.] kindly add ‘comparing with the control (0 µg/ml).
Response: Thanks for pointing this out. The relevant information has been added (Line 90).
- The discussion and references parts are fine and covered the aim and the finding of this article.
With my pleasure
Response: Thanks for your appreciation.
References:
- Ho, J.C.; Michalak, A.M.; Pahlevan, N. Widespread global increase in intense lake phytoplankton blooms since the 1980s. Nature 2019, 574(7780), 667-670.
- Huang, I.S.; Zimba, P.V. Cyanobacterial bioactive metabolites—A review of their chemistry and biology. Harmful Algae 2019, 83, 42-94.
